# The Microbiomes of Various Types of Abandoned Fallow Soils of South Taiga (Novgorod Region, Russian North-West)

Evgeny V. Abakumov [1,*], Grigory V. Gladkov [1,2], Anastasiia K. Kimeklis [1,2] and Evgeny E. Andronov [2,3]

1 Department of Applied Ecology, St. Petersburg State University, 199034 Saint-Petersburg, Russia; g.gladkov@spbu.ru (G.V.G.); a.kimeklis@spbu.ru (A.K.K.)

2 Laboratory of Microbiological Monitoring and Bioremediation of Soils, All-Russian Research Institute for Agricultural Microbiology, 196608 Saint-Petersburg, Russia; eeandr@gmail.com

3 V.V. Dokuchaev Soil Science Institute, 119017 Moscow, Russia

* Correspondence: e.abakumov@spbu.ru or e_abakumov@mail.ru

**Abstract:** More than 30 years have passed after the collapse of the Soviet Union, and huge areas of soil were left in a fallow state. The study of the microbiological status of fallow soils is an extremely urgent task because fallow soils represent the "hidden" food basket of Eurasia. In this context, we studied the influence of land use type (pasture, vegetable garden, hayfield, or secondary afforestation) on key agrochemical parameters and parameters of soil microbial biodiversity. All anthropogenically transformed soils included in the analysis showed increased humus content and pH shift to a more neutral side compared to the mature soil; the same seemed to be the case for all nutrient elements. It was established that the key factor regulating soil microbiome composition shift was the duration and degree of irreversibility of an agrogenic impact. The key phyla of soil microorganisms were Pseudomonadota, Acidobacteriota, Verrucomicrobiota, Bacteroidota, and Actinobacteriota. The proportion of other phyla was quite variative in soils of different land use. At the same time, all the 30-year-old abandoned soils were more similar to each other than to mature reference soil and 130-year-old soils of monoculture vegetable gardens. Thus, the first factor, regulating soil microbiome composition, is a continuation of soil agrogenic transformation. The second factor is the type of land use if the soil age was equal for fallow territory in the case of one initial podzol soil and one type of landscape. Thus, 30-year-old abandoned soils are intermediate in terms of microbial biodiversity between pristine natural podzols and plaggic podzol. It could be suggested that in the case of secondary involvement of soils in agriculture, the composition of the microbiome may turn to mature soil or to plaggic soil under intensive amelioration.

**Keywords:** abandoned soils; podzols; DNA sequencing; soil microbiome; soil fertility; south taiga

## 1. Introduction

Soil conversion to fallow in the post-Soviet period affected huge areas of land previously used for arable farming for hundreds of years because its history in Russia is very long. Fallow lands play an important role in the sequestration of carbon dioxide in soils. Abandoned post-agricultural soils may accumulate or release carbon, depending on the type and age of abandoning and the type of land use [1]. Even though Russia ranks first in the world in the availability of land resources and is among the top five in terms of arable land, currently up to 40 out of 120 million hectares have been extracted from agricultural practices. This land was converted to fallow (abandoned lands) and is being transformed by natural and anthropogenic processes. The easiest and least expensive way to increase productive arable land and thereby dramatically increase the agricultural potential of the country is to return these 40 million hectares into circulation [2,3]. The area of fallow land in Northwest Russia is not decreasing, while the share of arable land is characterized by a steady decline, leading to the development of different scenarios of succession and soil formation. The accumulation of organic matter in fallow soils and an increase in the

diversity of the molecular composition of humic acids in the profile vertical scale were observed during the postagrogenic successions. At the same time, a decrease in the share of aromatic components in the composition of organic matter is observed [4]. The last may lead to ambiguous effects on the emission of carbon dioxide.

Some of the soils of fallow lands, such as the outer islands of the Gulf of Finland, cannot be involved in the agricultural turnover and will remain untouched for a long time, as the islands are currently not inhabited [5]. These fallow agro-soils are unique natural monitoring models that can be used to study temporal trends in the soil-succession series. Some fallow lands will not be returned to the agricultural turnover because specially protected natural areas at regional and federal levels have been created on them. In the North-West of the Russian Federation, these are the Nizhnesvirsky State Reserve and the Valday National Park. The chronoseries of soils in such areas have been partially studied in terms of vegetation changes [6] and soil microbiome [7]. At the same time, the investigation of soil microbiome for benchmark, current agricultural, and fallow soils are few across the whole world and the Russian part of Eurasia [8]. Fragmentary data are available on the microbiome of Russian soils, including agrogenic soils [9–11], but these data are still insufficient to understand how the course of succession and land use affect the taxonomic composition and functional organization of the microbiome.

Since the soil microbiome is responsible for key biochemical processes in soils as well as their fertility, the study of the taxonomic composition of the microbiome is a priority for fallow soils. The relevance of this is reinforced by the recent interest in the re-engagement of agro-soils in agricultural practices. Studying the microbiome of fallow soils is also important in the context of elucidating their role in climate change and estimating carbon sequestration rates. Soils formed in the territory of the last glaciation as the youngest on the East European Plain can be one of the informative models for such studies. The end-moraine (glacial till) zone of the Valdai glaciation in the Novgorod region is the end zone of the distribution of the last glaciation [12]. Moraines, including local ones, zanders, and fluvioglacial sands, as well as ancient Pra-Msta river alluvial deposits, overlain by thin rewashed glacial deposits, are widespread here. The history of land use and agricultural practice here is well known and documented, as soils became abandoned from an arable form in 1993; thus, we have a good opportunity to compare various ways of abandoned soil development: afforestation of 30 and 70 years, pasturing, growing of meadow vegetation, current and abandoned vegetable gardens with one type of initial soil—podzol in fine sands, sublayered by the moraine loams of the Valday age.

That is why this work aimed to analyze the diversification of soil microbiome under the effect of different land use on the same type of soil and parent material in the southeastern part of the North-West Russian plain (Valday-Krestsy last glaciation moraine line). The following objectives were formulated: (1) to analyze soil morphology and chemical characteristics of the fine earth; (2) to investigate the taxonomy composition of soil microbiome; and (3) to analyze the parameters of alpha- and beta biodiversity of soil microorganisms in relation with type of land use and soil chemical characteristics.

## 2. Materials and Methods

### 2.1. The Study Sites

The study site's location is given in Figure 1. The research site is located in the territory of North-West Russia, in the Borovichi district of Novgorod region, in the vicinity of Velikiy Porog village. The sites under study are no more than 700 m apart, all of which is a single agro-landscape of the former state farm "Opechensky", which ceased to exist after the collapse of the Soviet Union.

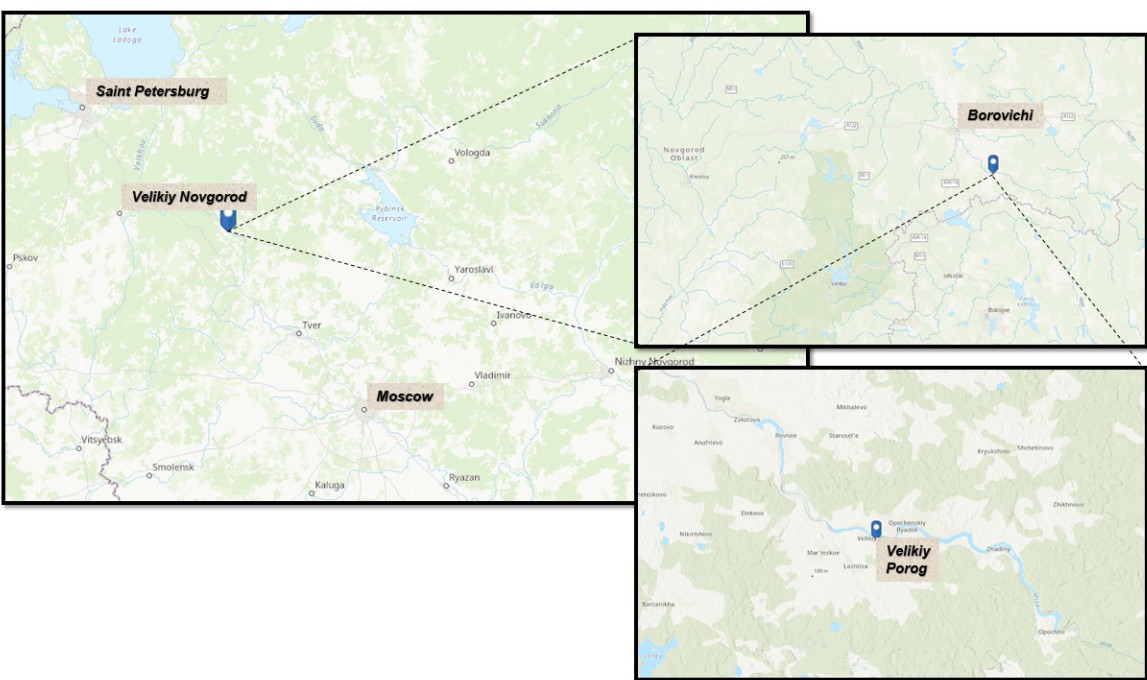

**Figure 1.** The location of the study area. Soil samples were collected near the Velikiy Porog village in the Borovichi district of Novgorod region (North-West Russia).

The average precipitation amount is 587 mm per year with evaporation about 430 mm. The annual mean temperature is 4.3 °C. The duration of the frost-free period is up to 130 days per year. The plot belongs to the south taiga bioclimatic zone with domination of podzols and retisols on drained watersheds and histic/gleyic soils in overmoisted landscape positions.

The history of farming in this region dates back a few centuries, but in this case, we studied the objects of 30-year-old fallow fields. The lands became fallow immediately after the collapse of the Soviet Union, i.e., in 1993 they were no longer plowed and fertilized. Further, a part of the fallow land was overgrown with secondary forest, a part was used continuously for haying, a part was subjected to permanent pasture digression (grazing of small cattle), and a part was used as a vegetable garden. Thus, the divergence of the soil formation process during 30 years after the land transition to fallow land is possible. We also studied the soil that reliably existed in arable condition for more than 130 years—a thick plaggen soil on podzol. All studied soils were formed on one type of parent material: sandy loam of water-glacial origin, underlain at the depth of 70–80 cm by red-brown moraine loams. The soil profiles are given in Figure 2. The study sites are located on the marginal part of the last, Valdai (Vyurm) glaciation, in the southern taiga subzone.

The natural (benchmark) vegetation of the sampling plots is Norway Spruce forest stands. The area of plowed land in this area during the Soviet era exceeded 80%, and even those spruce forests that appear to be primary underwent very significant anthropogenic impact. The sample plots, used for the analysis, are described in Figure 2. The mature soil (Figure 2a) was a podzol with a well-expressed iron-illuviated horizon, sublayed by a gleyification layer—G. Soils of 30-year old vegetable gardens (Figure 2b), secondary forest (Figure 2c), and pasture digression (Figure 2d) have loosed initial E and BF horizons; thus, they were involved to the arable horizon during Soviet times, and show no signs of secondary podzolization during the subsequent thirty-year stay in the deposit. Only the hayfield soil (Figure 2e) shows some signs of secondary podzolization in the upper part of the old plow horizon. As for the oldest arable soil, plaggen (Figure 2f), that has a humus A horizon with a total depth of about 40 cm, these soils were used for the monoculture of potatoes for at least 130 years. The coordinates of plots are the following: a—58.265268 N,

34.091997 E, b—58.269883 N, 34.083628 E, c—58.268745 N, 34.085468 E, d—58.269535 N, 34.083580 E, e—58.268671 N, 34.083279 E, and f—58.270034 N, 34.081949 E.

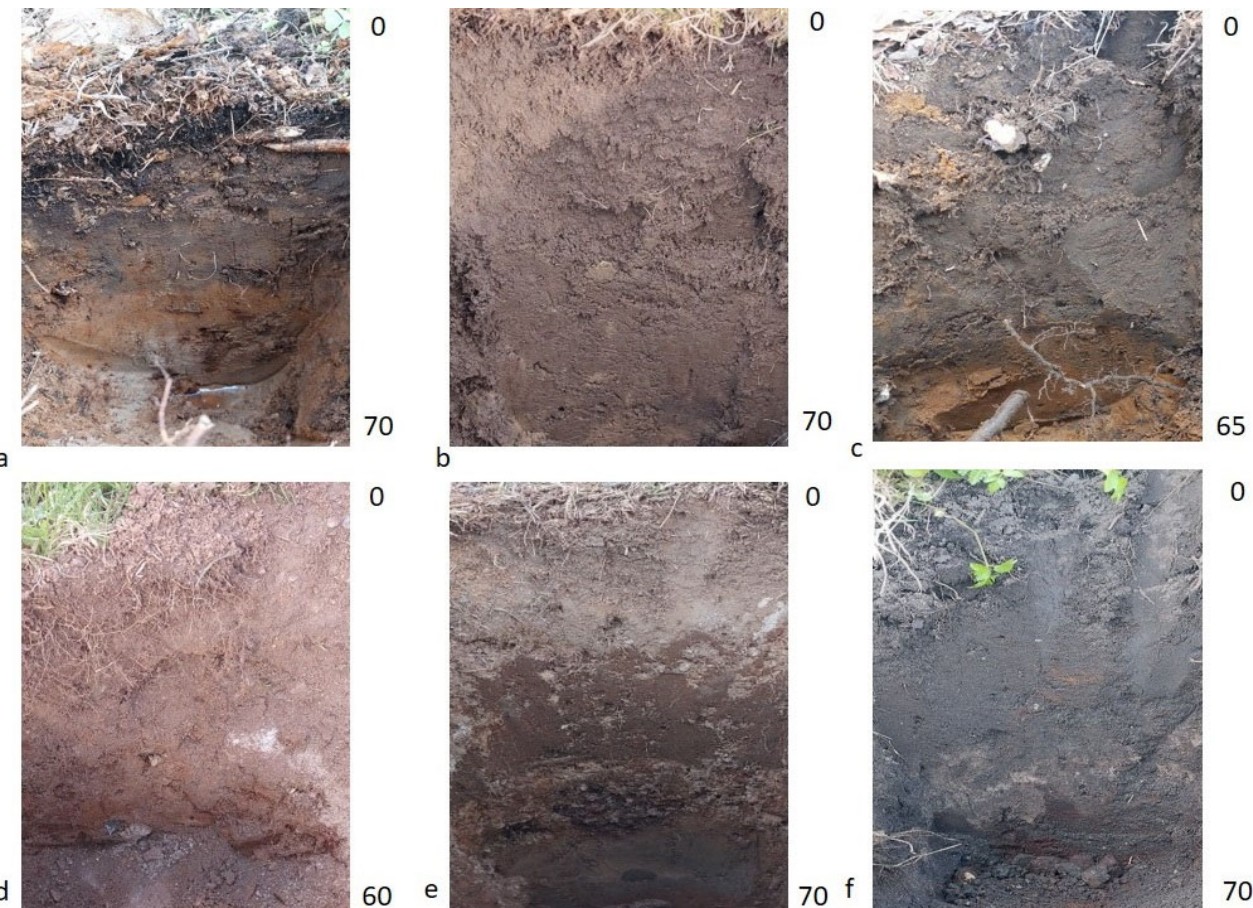

**Figure 2.** The studied soil profiles. (**a**) Mature soil (Mature soil), (**b**) 30-year-old abandoned vegetable garden (Garden), (**c**) 30-year-old secondary forest (Forest), (**d**) 30-year-old pasture digression (Pasture), (**e**) 30-year-old haymaking field (Hayfield), (**f**) 130-year-old plaggen (Plaggen) (soils formed under the long-term fertilization of bedding manure material applied in spring and fall from the farm. At the same time, horse, cow, and sheep manure was applied until 1940, and cow and sheep manure until 1970, after which chicken and goat manure dominated). Numbers to the right of the photos indicate the soil profile depth in cm.

*2.2. Laboratory Analysis*

The soil samples were collected on 10 May 2023 from the soil section wall at a depth of 10 cm using a special soil knife. The soil samples were taken in triplicate in each location. For the topsoil horizons, an average sample weighing 500 g was formed for routine soil analyses. For microbiological analysis, 2 g samples were taken in sterile tubes (in 4 replications). The samples for microbiological analysis were transported at +4 °C and stored at −20 °C.

For agrochemical analyses, the samples were air-dried, ground, and passed through a 2 mm sieve. The pH of the soil solution was measured using a Milwaukee Mi106 (Milwaukee Electronics (USA)) pH meter. A soil solution was prepared in a ratio of 1:2.5 with water [13]. The soil organic carbon (SOC) content was determined via the Tyurin method (comparable to Walkey and Blake's procedure) based on the oxidation of soil organic matter with a mixture of potassium dichromate and concentrated sulfuric acid [14]. The basal respiration (BR) and substrate-induced respiration (SIR) were measured according to the standard procedure in a laboratory in closed chambers under 60% soil humidity of whole water capacity. The content of available forms of ammonium nitrogen (N-NH$_4^+$)

and nitrate nitrogen (N-NO$_3^-$) was determined using a potassium chloride solution. The amounts of free potassium and phosphorus were determined via the Kirsanov method using phosphorus and potassium extraction with a 0.2 N hydrochloric acid solution [15,16]. The particle size distribution was measured using the classical sedimentometry method in 1 L volume cylinders with gravimetrical counting of each particle size fraction.

The total soil DNA was isolated in four replicates for each soil site by using the MN NucleoSpin Soil Kit (Macherey-Nagel, Dueren, Germany) using a Precellys 24 homogenizer (Bertin, Montigny-le-Bretonneux, France) according to the manufacturer's protocol. Quality control was carried out via PCR and agarose gel electrophoresis. The paired-end sequencing of the V4 variable region of the 16 S rRNA gene for a total of 24 samples was performed on the Illumina MiSEQ platform (Illumina, San Diego, CA, USA), using the primers 515 f (GTGCCAGCMGCCGCGGTAA) and 806 r (GGACTACVSGGGTATCTAAT) [17].

### 2.3. Data Processing and Statistical Analysis

The general processing of sequences was carried out in R 4.3 [18] using dada2 v. 1.28 (trimming at 200/180, error rate 2.5, truncation quality score 2) [19] and phyloseq v. 1.42 [20] packages, as described earlier [21]. The taxonomic annotation was performed with the SILVA 138.1 database [22] used as the training set; the phyla names were corrected according to LPSN [23]. The alpha-diversity (observed ASV and Simpson indices), beta-diversity (non-metric multidimensional scaling (NMDS) ordination of Bray-Curtis distances) metrics, and canonical correspondence analysis (CCA) were calculated using vegan v. 2.6-4 package [24] and visualized in phyloseq. The PERMANOVA analysis [25] was carried out using the adonis2 function in the vegan package.

## 3. Results and Discussion

Data on the soil chemical composition are given in Table 1. All the investigated soils were characterized by slightly acid pH values. The mature soil was more acidic than agrogenic ones. The highest pH was in plaggen soils; this is because the local population uses ash from stove heating to fertilize soils annually, as well as periodically applying purchased lime fertilizers. Thus, 30 years of being in the fallow state was not enough for a sharp change in the acid-base balance of soils. These data well correspond to the recently published results [26], which demonstrate that only after 25–30 years of a fallow state, a rapid change in soil acidity was detected. Thus, during the first three decades, abandoned soils act as a buffer in terms of acidity even in the case of cessation of fertilization, when the soil may not acidify for a long time, although reverse trends are possible [2]. According to the measurements of basal respiration, the respiration rate was the lowest in the mature soil and the highest in the secondary forest. T well corresponds with the previous data that soil organic matter is more stabilized in the end stage of pedogenesis, while the mineralization of a huge amount of fresh litter organic material may take place in the stage of active ecogenesis and soil formation [27]. The lowest values of *p* and K were in the mature podzol soils, which is typical for this type of soil with pronounced eluviation of any cations [28]. The highest level of these elements' accumulation was detected in the long-term amended plaggen soils. Other soils of our experiment were in the intermediate position in terms of *p* and K concentration between mature and plaggen soils. The pasture soil was the only one to demonstrate some increase in the mentioned nutrients, which could be a result of soil fertilization due to grazing livestock manure, or high spatial variability of soil agrochemical state, typical for the soil of post-Soviet agroecosystems [28,29] The content of ammonia nitrogen was low and that nitrate nitrogen was very low [30] in all soils investigated, with some variations among land use type. Thus, data on soil agrochemical composition may play a role as a predictor of soil microbial community composition and dynamics, which are analyzed below.



**Table 1.** Soil chemical analyses.

| Sample ID | pH | TOC, % | BR | SIR | N-NH$_4^+$ | N-NO$_3^-$ | $p$ | K |
|---|---|---|---|---|---|---|---|---|
| | | | μg CO$_2^-$C/g × hour | | Exchangeable forms, mg kg$^{-1}$ | | | |
| Mature Soil | 5.73 | 0.83 ± 0.06 | 0.82 ± 0.05 | 2.88 ± 0.08 | 4.08 ± 0.24 | 0.27 ± 0.09 | 25 ± 5 | 10 ± 2 |
| Garden | 6.11 | 1.40 ± 0.06 | 0.50 ± 0.09 | 0.87 ± 0.03 | 6.49 ± 0.13 | 0.49 ± 0.05 | 59 ± 4 | 13 ± 3 |
| Forest | 6.67 | 2.55 ± 0.05 | 2.03 ± 0.05 | 7.31 ± 0.22 | 11.78 ± 0.22 | 1.06 ± 0.08 | 70 ± 3 | 69 ± 4 |
| Pasture | 6.37 | 2.42 ± 0.09 | 1.60 ± 0.15 | 8.17 ± 0.72 | 11.52 ± 0.24 | 2.31 ± 0.09 | 161 ± 7 | 195 ± 6 |
| Hayfield | 6.23 | 2.21 ± 0.07 | 1.45 ± 0.12 | 3.48 ± 0.09 | 7.91 ± 0.31 | 0.70 ± 0.06 | 93 ± 6 | 93 ± 8 |
| Plaggen | 6.87 | 3.77 ± 0.03 | 1.27 ± 0.02 | 4.93 ± 0.25 | 5.92 ± 0.16 | 3.61 ± 0.14 | 499 ± 12 | 505 ± 19 |

Data on soil particle size distribution are given in Table 2. All the soils investigated are classified as sandy-textured with the dominance of the fraction of coarse sand inherited from the sandy-textured parent material. This is quite typical for podzol soils [31]. The physical clay particles were very low, which may predict the low absorption ability of soils both for chemical compounds and for microorganisms [32]; nevertheless, this influence was shown as not so direct and driven by particle size, especially in the case of the fungal community [33].

**Table 2.** Particle size distribution, fraction content, to fine earth (diameter of granulometric fractions).

| Sample ID | 1–0.25 | 0.25–0.05 | 0.05–0.01 | 0.01–0.005 | 0.005–0.001 | <0.001 | <0.01 |
|---|---|---|---|---|---|---|---|
| Mature soil | 50 | 38 | 3 | 1 | 1 | 7 | 9 |
| Garden | 45 | 30 | 16 | 1 | 1 | 7 | 9 |
| Forest | 29 | 48 | 11 | 1 | 4 | 7 | 12 |
| Pasture | 40 | 46 | 4 | 3 | 4 | 3 | 10 |
| Hayfield | 43 | 42 | 6 | 1 | 3 | 5 | 9 |
| Plaggen | 55 | 28 | 5 | 2 | 4 | 6 | 12 |

*Soil Microbiome*

Libraries of the 16 S rRNA gene for five sites of abandoned soils and one site of control soil were obtained. A total of 428,559 reads were classified into 5115 phylotypes. All investigated plots differed in their composition and representation of microorganisms. The most pronounced differences were detected between fallow soils, plaggen soils, and mature soil.

The parameters of alpha diversity of soil microorganisms are given in Figure 3. The number of phylotypes (ASVs) was different among the sites investigated. The lowest values of observed ASV, indicating community richness, were detected in the microbiomes of both forest soils—mature and 30-year-old forest on fallow land. The opposite data were obtained for the podzol soil microbiome under the urbanization effect [34]. It was previously shown that the composition of the microbiome of podzols was strongly influenced by the duration of the soil in a disturbed state [35]; in our case, several objects had one age of accumulation—30 years, which means that the microbiome is determined not by the age of the soil, but by the type of its use. The Shannon index characterizes evenness in the community structure and shows the degree of dominance of certain species in the community structure [36]. The lowest value of this index was in the mature forest soil and plaggen (Figure 3). While the richness of the plaggen microbiome did not reliably differ from fallow soils, its Simpson index was significantly lower than that of other fallow soils, which suggests that there was a decrease in evenness under long-term soil use for potato planting during more than 130 years. The values of all alpha indices were the highest in the soil under the pasture digression; thus, we can suggest that high soil biodiversity may be a result of intensive soil degradation. In addition, the soil of the hayfield had increased Shannon and Simpson indices values. Thus, both ecosystems of grassland were characterized by increased biodiversity. To conclude, forest ecosystems had less developed microbial

biodiversity, but in the case of pasture, hayfield, garden, and plaggen soils, there was an increase in the diversity of the soil microbiome.

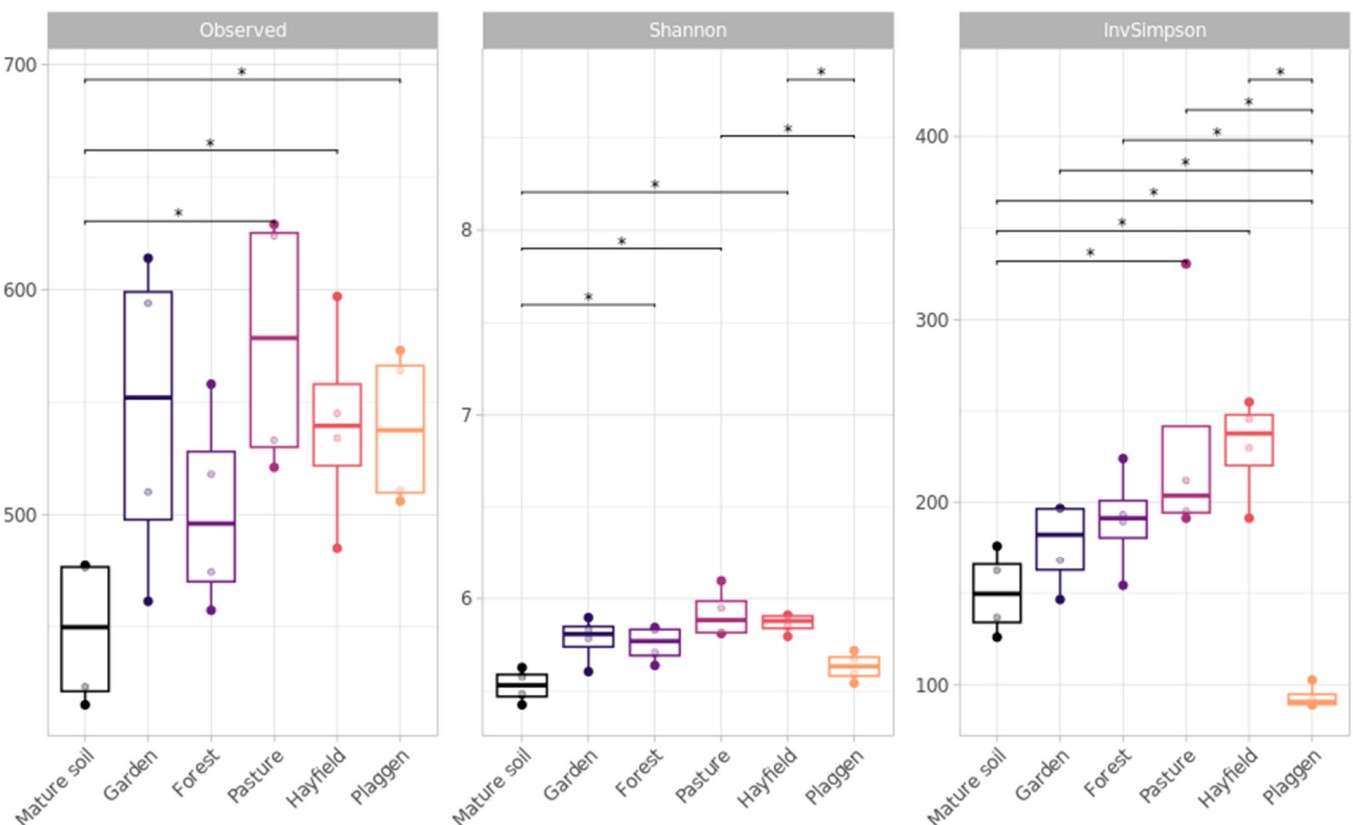

**Figure 3.** Parameters of alpha diversity (observed, Shannon, InvSimpson) of the microbial communities, each sample site in four replicates. Significant differences above the boxplots were assessed using the Mann–Whitney test, (*) *p*-value < 0.05.

An analysis of beta-biodiversity parameters (Figure 4) showed that the background soil and 130-year-old plaggen were the most different from each other and other fallow soils in terms of microbiome composition. In this case, the longest history of soil development in the course of agricultural impact was observed. In general, time is a very serious factor in determining the composition and function of the soil microbiome [36,37].

Data on the CCA plot (Figure 5) demonstrated that microbiomes of plaggen and mature soil reacted very differently to chemical composition than fallow soils. For plaggen soils, the increased amounts of available phosphorus, potassium, and nitrate nitrogen were the most important factors for explaining microbiome diversification. Total organic carbon and pH also contributed to the specificity of the plaggen microbiome. The basal and substrate-induced respiration values as well as ammonium nitrogen were the most important factors in the clusterization of all abandoned soils (hayfield, pasture, secondary forest, vegetable garden). Consequently, the mature soil microbiome was characterized by a negative correlation with the level of soil respiration and ammonium nitrogen content, which may demonstrate a stable state in mature soil in terms of biochemistry and microorganism activity in comparison with other ones.

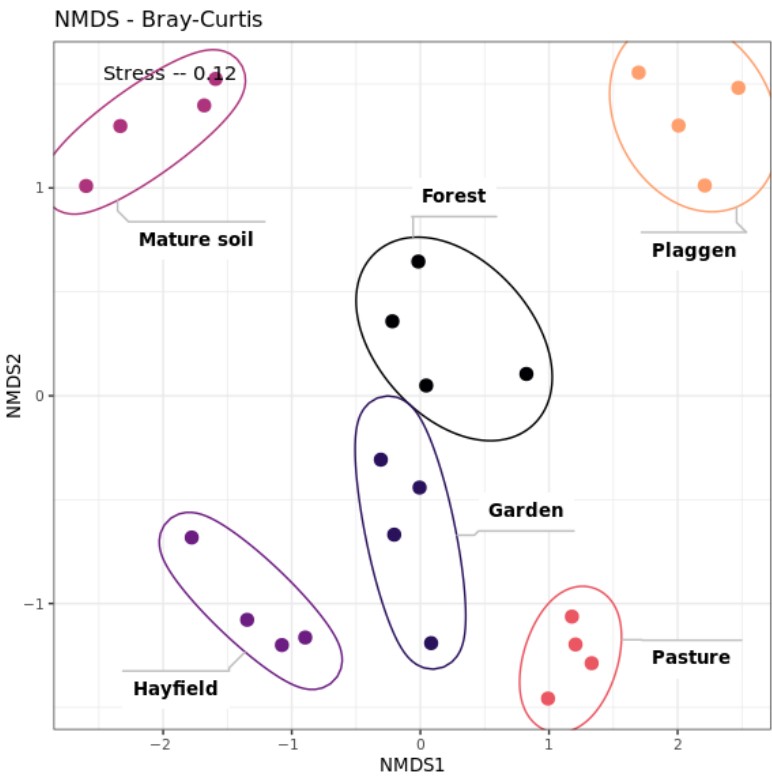

**Figure 4.** NMDS-plot of beta-biodiversity (Bray–Curtis distances), which demonstrates local differences between samples of abandoned (Plaggen, Garden, Pasture, Forest, Hayfield) and control (Mature soil) soils. Four replicate samples from one site are surrounded by ellipses.

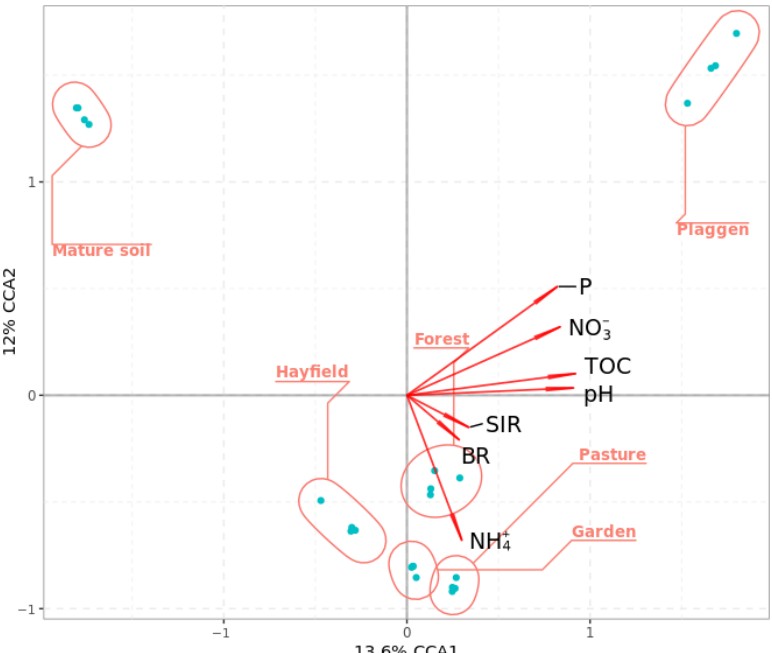

**Figure 5.** CCA plot demonstrating linear relationships between chemical characteristics and soil microbial communities of different abandoned soils. TOC—total organic carbon, P—phosphorus, $NH_4^+$—ammonium, $NO_3^-$—nitrates, SIR—soil-induced respiration, BR—basal respiration. K (potassium) is aliased with P because they were collinear (redundant). Percentages on the axis are the proportion explained values.

The relative abundances of major phyla are provided in Figure 6. The most represented phylotypes belonged to Pseudomonadota, Acidobacteriota, Verrucomicrobiota, Bacteroidota, and Actinobacteriota. All studied soils in terms of microbiome composition can be characterized as poorly anthropogenically transformed developed soils. This was indicated by a large number of representatives from the slightly cultured phyla Acidobacteriota and Verrucomicrobiota and bacteria that are more characteristic of the complex metabolically autochthonous microbiome (representatives of Planctomycetota, Chloroflexota, *Puia*, and *Haliangium*) (Figure 7). This well corresponds with the data of another fallow soil experiment, which show that in a 25-year-old fallow plot of albic retisol, the intensity of microbiological processes was close to those in virgin soddy-podzolic soils (Retisols), which were used as a control [38].

| | Mature soil | Garden | Forest | Pasture | Hayfield | Plaggen |
|---|---|---|---|---|---|---|
| Acidobacteriota | 32.9 | 24.3 | 21.5 | 18.2 | 25.6 | 14.3 |
| Gammaproteobacteria | 11.1 | 12.7 | 16.9 | 12 | 9.7 | 14 |
| Alphaproteobacteria | 18 | 9.5 | 13.5 | 8.6 | 11.9 | 5.6 |
| Verrucomicrobiota | 7 | 13.1 | 11.1 | 13 | 15.1 | 5.5 |
| Bacteroidota | 4.6 | 8.3 | 9.4 | 15.3 | 6.8 | 10.7 |
| Actinobacteriota | 8.4 | 4.7 | 6.4 | 7.6 | 5.5 | 4.2 |
| Bacillota | 1.2 | 7.9 | 4.3 | 6.3 | 3.8 | 8.8 |
| Planctomycetota | 2.5 | 5.6 | 3.4 | 6.9 | 7.6 | 5.7 |
| Chloroflexota | 7.4 | 3.2 | 3.1 | 2 | 6 | 3.5 |
| Thermoproteota | 0.4 | 0.2 | 0.6 | 1.7 | 0.1 | 13.7 |
| Myxococcota | 0.9 | 2.7 | 1.5 | 3.1 | 2.1 | 2.2 |
| Patescibacteria | 1.3 | 1.2 | 1.4 | 1.2 | 1.5 | 1.2 |
| Gemmatimonadota | 0.5 | 0.4 | 2 | 1 | 1 | 2 |
| Nitrospirota | 0 | 0.2 | 1.7 | 0.4 | 0 | 3.5 |
| Methylomirabilota | 0.1 | 1.6 | 0.9 | 0.1 | 0.1 | 0.6 |
| Remaining taxa (51) | 3.5 | 4.5 | 2.3 | 2.8 | 3 | 4.4 |

**Figure 6.** The relative abundance of major phyla in the microbiomes of abandoned and background soils. Pseudomonadota members are presented by two major classes. A darker color shows higher values, and a lighter one, lower.

The dominance of Acidobacteriota is known to be typical for organogenic and organomineral soil horizons [34]. Also, the dominance of Alpha- and Gammaproteobacteria, Actinobacteriota, Verrucomicrobiota, and Planctomycetota is known as typical for superficial podzol soil organic and organo-mineral horizons with acid reaction [8,37]. The representatives of Chloroflexota were more abundant in the mature soil, which may be linked to the high amount of fresh organic matter in the developed forest ecosystem, as it was mentioned recently that this group of microorganisms plays an important role in organic substance decomposition [39]. The microbiome of natural soils combines with the microbiome of agrogenic soils at both taxonomic and functional levels. Thus, the microbiome has components of the microbiome of sandy podzols and different variations of agrogenic soils. Thus, there was an evident trend in the increasing of Bacillota in all anthropogenically affected soils in comparison with the mature one. This is consistent with the findings that Bacillota appeared after the initial stage of lignocellulose decomposition [40]. The highest portion of Thermoproteota was in plaggen soils, which well corresponds with data on a high content of this group of microorganisms in fertile black soils of central Europe [41]. The plaggen soil was similar to black soils in terms of humus content, but not in terms of soil acidity; thus, the organic matter content may be a key factor of specificity of the soil microbiome of old cultivated soil. The phylum (Thermoproteota) is also known as a predictor of soil aggregation [41], and the aggregation intensively depends

on organic matter content. The portion of Myxococcota was increased in all altered soil in comparison with benchmark (initial) one. The representatives of the Myxococcota phylum are well-known as micro predators able to destroy bacteria and eukaryotic organisms as well as to degrade complex macromolecules. An increase in their share in agricultural soils compared to natural soil may indicate certain shifts in soil metabolism [42] and may be responsible for the intensification of carbon-containing substance turnover. The role of Paterscibateria in soils is linked with the transport of metal in soil [43]; thus, its portion was more or less equal in all soils investigated because they belong to one geochemical group of sandy-textured iron-enriched south-taiga soils. High levels of nitrifying Nitrososphaerota archaea and Nitrospirota representatives were present, indicating that the microbiome reacted sharply to the introduction of excess nitrogen into the soil. These data contribute to the hypothesis [34] that the age of soil disturbance may play the most important role in the diversification of soil microbial biodiversity. The representatives of Gemmatimonadota did not show any significant differences among the soils.

| | Mature soil | Garden | Forest | Pasture | Hayfield | Plaggen |
|---|---|---|---|---|---|---|
| Verrucomicrobiota; Candidatus Udaeobacter | 2.8 | 5.8 | 6.8 | 7.2 | 10 | 2.4 |
| Acidobacteriota; Candidatus Solibacter | 2.1 | 4.3 | 2.3 | 2.6 | 3.6 | 0.2 |
| Bacillota; o__Bacillales_Seq2 | 0.3 | 3.9 | 2 | 1.5 | 2.1 | 3.2 |
| Acidobacteriota; Bryobacter | 2 | 2.7 | 1.5 | 2.2 | 3.3 | 0.6 |
| Acidobacteriota; RB41 | 0 | 1.6 | 1.9 | 2.8 | 1.6 | 2.8 |
| Bacteroidota; Puia | 3.2 | 1.2 | 1.5 | 1.1 | 2.3 | 0 |
| Pseudomonadota; Acidibacter | 3.3 | 0.8 | 0.8 | 1.7 | 1.3 | 0.8 |
| Thermoproteota; f__Nitrososphaeraceae_Seq9 | 0 | 0 | 0 | 0 | 0 | 7.5 |
| Pseudomonadota; Pseudolabrys | 1.7 | 1.2 | 1.3 | 1.4 | 1 | 0.3 |
| Bacteroidota; Flavobacterium | 0 | 1.1 | 1 | 2.1 | 0.3 | 2.3 |
| Pseudomonadota; mle1-7 | 0 | 1.8 | 3.1 | 0.3 | 0.1 | 1.2 |
| Bacillota; o__Bacillales_Seq11 | 0.1 | 1.5 | 0.6 | 3 | 0.3 | 0.8 |
| Pseudomonadota; Bradyrhizobium | 1.3 | 1 | 1.3 | 1.1 | 1.2 | 0.2 |
| Verrucomicrobiota; Candidatus Xiphinematobacter | 0.9 | 2.4 | 1.4 | 0.4 | 0.7 | 0.2 |
| Pseudomonadota; f__Xanthobacteraceae_Seq4 | 2 | 0.4 | 1.6 | 0.6 | 1 | 0 |
| Nitrospirota; Nitrospira | 0 | 0.1 | 1.7 | 0.4 | 0 | 3.5 |
| Actinobacteriota; Acidothermus | 1.4 | 1.1 | 0.5 | 0.5 | 1.2 | 0 |
| Chloroflexota; HSB OF53-F07 | 0.4 | 1.8 | 0.1 | 0.1 | 2 | 0.1 |
| Pseudomonadota; Ellin6067 | 0 | 0.6 | 1.2 | 1.1 | 0.7 | 0.7 |
| Verrucomicrobiota; Chthoniobacter | 0.4 | 0.4 | 0.8 | 1.3 | 0.9 | 0.4 |
| Myxococcota; Haliangium | 0.1 | 0.9 | 0.4 | 0.9 | 1.2 | 0.2 |
| Acidobacteriota; JGI 0001001-H03 | 0 | 0.2 | 0.7 | 1.4 | 0.9 | 0.3 |
| Verrucomicrobiota; ADurb.Bin063-1 | 0.1 | 1.3 | 0.3 | 1 | 0.7 | 0 |
| Bacillota; Paenisporosarcina | 0 | 0.8 | 0.5 | 0.4 | 0.6 | 1.2 |
| Acidobacteriota; o__Subgroup 2_Seq34 | 0.2 | 2.1 | 0.8 | 0 | 0.3 | 0 |
| Bacteroidota; f__Microscillaceae_Seq33 | 0 | 0.1 | 3 | 0 | 0.1 | 0.1 |
| Pseudomonadota; MND1 | 0 | 0.4 | 0.9 | 0.4 | 0.2 | 1.2 |
| Acidobacteriota; o__Acidobacteriales_Seq24 | 1.5 | 0.2 | 0.4 | 0 | 0.5 | 0.2 |
| Actinobacteriota; c__Acidimicrobiia_Seq14 | 2.9 | 0 | 0 | 0 | 0 | 0 |
| Planctomycetota; Pir4 lineage | 0 | 0.3 | 0.3 | 0.6 | 0.4 | 1.2 |
| Bacteroidota; OLB12 | 0 | 0.7 | 0.8 | 0.5 | 0.7 | 0 |
| Bacteroidota; Ferruginibacter | 0 | 0.3 | 0.3 | 1.4 | 0.2 | 0.4 |
| Pseudomonadota; o__WD260_Seq29 | 1.5 | 0.1 | 0.6 | 0.1 | 0.3 | 0 |
| Pseudomonadota; GOUTA6 | 0 | 1.1 | 0.7 | 0.3 | 0 | 0.3 |
| Pseudomonadota; Burkholderia-Caballeronia-Paraburkholderia | 1.3 | 0 | 0.8 | 0 | 0.3 | 0 |
| Pseudomonadota; Hyphomicrobium | 0.3 | 0.1 | 0.7 | 0.3 | 0.1 | 0.9 |
| Bacteroidota; Terrimonas | 0 | 0.6 | 0.3 | 0.6 | 0.2 | 0.7 |
| Pseudomonadota; IS-44 | 0 | 0 | 1.3 | 0.2 | 0 | 0.7 |
| Pseudomonadota; Pseudomonas | 0.1 | 0.1 | 0.1 | 0.5 | 0.2 | 1.4 |
| Pseudomonadota; f__Xanthobacteraceae_Seq66 | 0 | 0.6 | 0.4 | 0.7 | 0.5 | 0 |
| Remaining taxa (3626) | 69.9 | 56.4 | 55.3 | 59.1 | 59.2 | 63.9 |

**Figure 7.** The relative abundance of the separate phylotypes in the microbiomes of abandoned and background soils. The row name includes phylum and the lowest determined taxon. A darker color shows higher values, and a lighter one, lower.

## 4. Conclusions

For the first time for the southern taiga subzone of the European territory of Russia, the development of soils and the formation of their microbiome 30 years after the transition to a fallow state were studied, as well as these parameters for the background soil and 130-year continuously used for potato cultivation plaggen podzol soil. All soils were formed from the same podzol formed on fluvioglacial sandy textured parent materials. Different fallow variants were studied: hayfield, vegetable garden, pasture digression, and secondary forest. It was found that the greatest differences in the chemical composition of soils and microbiome characteristics were characteristic of the pair of background podzol and 130-year-old plaggen. In the plaggen, the main factor affecting the microbiome was the content of organic matter, exchangeable phosphorus, and potassium compounds. The

microbiome composition of 30-year-old fallow soils of all listed variants differed from both background soils and plaggen. All studied anthropogenically transformed soils showed increased humus content and pH shift to a more neutral side compared to the mature soil. The same seemed to be the case for all nutrient elements. It was established that the key factor of soil microbiome composition shift was the duration and degree of irreversibility of an agrogenic impact. The second factor was the content of nutrition elements—ammonium nitrogen in younger fallow soils and phosphorus and potassium in longer-developed soils.

**Author Contributions:** E.V.A.: writing—review and editing, project administration; funding acquisition; A.K.K.: laboratory procedures, data curation, investigation; G.V.G.: data curation, investigation, investigation, statistical treatment; E.E.A.: writing—review and editing. All authors have read and agreed to the published version of the manuscript.

**Funding:** This research was supported by the Russian Science Foundation, project No. 23-16-20003, dated 20 April 2023, and the Saint Petersburg Scientific Foundation, agreement No. 23-16-20003, dated 5 May 2023.

**Institutional Review Board Statement:** Not applicable.

**Informed Consent Statement:** Not applicable.

**Data Availability Statement:** Data are available at the NCBI Project PRJNA1023167, BioSample accessions SAMN37642152-7.

**Acknowledgments:** We thank the Centre for Genomic Technologies, Proteomics and Cell Biology (ARRIAM, Russia) for performing the preparation and sequencing of 16 S rRNA gene libraries. The authors are grateful to Timur Nizamutdinov, Roman Dyachkovsky, and Vyacheslav Polyakov, researchers of the Department of Applied Ecology, for their assistance in performing general chemical analyses of soils. This article is dedicated to the 300th anniversary of St. Petersburg State University.

**Conflicts of Interest:** The authors declare no conflict of interest.

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
