# Peer review of "The Microbiomes of Various Types of Abandoned Fallow Soils of South Taiga (Novgorod Region, Russian North-West)"

_agronomy, doi:10.3390/agronomy13102592_

Round 1

Reviewer 1 Report

The manuscript submitted by Abakumov et al. to the agronomy journal investigates changes in soil properties and the microbial community within fallow soils that have been abandoned for varying durations. The research reveals an increase in humus content, a shift towards a more neutral pH, and alterations in several key nutrient parameters. Furthermore, the study highlights the impact of soil agrogenic transformations and land use types on the soil microbial community. This type of investigation can provide valuable insights for assessing the quality and evolution of abandoned fallow soils, which is essential for agricultural development. However, substantial improvements are needed in the manuscript to meet the publication standards.

Major comments:

One limitation to address is the language used in the manuscript, which requires significant improvement. Efforts should be made to simplify sentences and avoid overly colloquial expressions, such as 'quiet ... (e.g., line 223, 248...)' or 'it is possible to speak about... (line 97).' Additionally, please ensure consistency in the use of tense, particularly when describing results observed or published in other studies.

The Introduction should be enhanced to highlight its key strengths more effectively. The extensive paragraph devoted to agricultural history could be streamlined, while the discussion of the necessity and current state of knowledge regarding the microbiome in abandoned fallow soils could be strengthened.

Thirdly, the amalgamation of results and discussion poses a challenge in maintaining a clear and focused narrative without being sidetracked by minor details. It is unclear whether this is a journal requirement or a preference of the authors. A clearer structure and a more focused narrative could be achieved if the results and discussion were presented as separate sections.

Minor comments:

Line 14, … was studied, should be ‘were studied’ as the subject is plural.

Line 17, key factor regulating soil microbime…

Line 20, phyla is already plural

Line 26. In could be suggested

Line 27-28, two time of ‘in case of’, maybe replace the 2nd one with ‘under’

Line33, ‘even thousands of years’ sounds a bit of over-confident for agriculture in that area, you’d better to provide citation if it is true, or tune down for a prudent consideration.

In Section 2.2, Laboratory Analysis, it would be beneficial to provide additional details regarding the library preparation. Specifically, could you clarify whether the PCR products from triplicates were pooled into a single library for sequencing or if each PCR product was sequenced separately? Additionally, it would be helpful to know whether you used paired-end or single-end sequencing on the Illumina Miseq platform, and whether the sequencing service was outsourced to a commercial company or conducted within your institute. Furthermore, could you please specify the parameters used in Dada2, including error rate values, sequence length and quality score cutoffs? Additionally, did you rarefy your data prior to calculating the alpha diversity indices?

On Line 153, it would be helpful to provide a numeric citation index for the vegan package. The indices mentioned from Line 147 to 150 do not align with the reference numbers. For Lines 151 to 155, the descriptions for NMDs and CCA seem somewhat repetitive, and it would be beneficial to condense them for clarity.

Line160, a typo, should be ‘slightly acidic pH’

Line 201-220, I would suggest to tune down the interpretation of Shannon index.

Starting from Line 228 onwards, I recommend reviewing the journal's italicization requirements for scientific names. Typically, scientific names up to and including the family level are italicized, while names beyond that level are not. Furthermore, consider that the discussion related to various phyla tends to divert from the primary storyline you intend to emphasize. It may be beneficial to restructure this section based on the points highlighted in Lines 273-274 to maintain better focus.

Regarding Figure 3, kindly include the number of observations for each boxplot. Based on the available information, it appears that the number of observations is limited. If this is the case, statistical significance may not be robust, and it may be advisable to omit it.

As for Figure 5, please ensure that the sum of contributions from each axis to the total variation, which can be explained by the environmental variables, does not exceed 100%. Please review your source code for troubleshooting.

Figure 6, class level (e.g. Alpha- and Gamma-proteobacteria) and phylum level were mixed. If this is what you want to demonstrate, please state in the figure caption.

Figure 7, mixed condition similar to Fig 6, as your mentioned at genus level, while a lot of were at taxonomic rank of order.

Last, ‘data availability’ about your sequencing data are missing. Typically, sequencing data is uploaded to databases like SRA of NCBI or ENA, and corresponding BioProject ID and sample IDs are provided.

English should be substentially improved.

Author Response

Reviewer 1

The manuscript submitted by Abakumov et al. to the agronomy journal investigates changes in soil properties and the microbial community within fallow soils that have been abandoned for varying durations. The research reveals an increase in humus content, a shift towards a more neutral pH, and alterations in several key nutrient parameters. Furthermore, the study highlights the impact of soil agrogenic transformations and land use types on the soil microbial community. This type of investigation can provide valuable insights for assessing the quality and evolution of abandoned fallow soils, which is essential for agricultural development. However, substantial improvements are needed in the manuscript to meet the publication standards.

We want to thank the reviewer for the high evaluation of the manuscript and the substantial work put into its reviewing. Especially we are very grateful for the comments concerning data processing and presentation.

Major comments:

One limitation to address is the language used in the manuscript, which requires significant improvement. Efforts should be made to simplify sentences and avoid overly colloquial expressions, such as 'quiet ... (e.g., line 223, 248...)' or 'it is possible to speak about... (line 97).' Additionally, please ensure consistency in the use of tense, particularly when describing results observed or published in other studies.

Thank you, the language quality has been improved

The Introduction should be enhanced to highlight its key strengths more effectively. The extensive paragraph devoted to agricultural history could be streamlined, while the discussion of the necessity and current state of knowledge regarding the microbiome in abandoned fallow soils could be strengthened.

Thank you, few paragraphs has beed added.

Thirdly, the amalgamation of results and discussion poses a challenge in maintaining a clear and focused narrative without being sidetracked by minor details. It is unclear whether this is a journal requirement or a preference of the authors. A clearer structure and a more focused narrative could be achieved if the results and discussion were presented as separate sections.

Thank you for this comment. The journal's website says that the Results and Discussion sections can be combined, which we have done, but we have changed the structure of the text.

Minor comments:

Line 14, … was studied, should be ‘were studied’ as the subject is plural.

Thank you, has been corrected.

Line 17, key factor regulating soil microbime…

Thank you, has been corrected.

Line 20, phyla is already plural

Thank you, has been corrected.

Line 26. In could be suggested

Thank you, has been corrected.

Line 27-28, two time of ‘in case of’, maybe replace the 2nd one with ‘under’

Thank you, has been corrected.

Line33, ‘even thousands of years’ sounds a bit of over-confident for agriculture in that area, you’d better to provide citation if it is true, or tune down for a prudent consideration.

Thank you, the text has been corrected.

In Section 2.2, Laboratory Analysis, it would be beneficial to provide additional details regarding the library preparation. Specifically, could you clarify whether the PCR products from triplicates were pooled into a single library for sequencing or if each PCR product was sequenced separately? Additionally, it would be helpful to know whether you used paired-end or single-end sequencing on the Illumina Miseq platform, and whether the sequencing service was outsourced to a commercial company or conducted within your institute. Furthermore, could you please specify the parameters used in Dada2, including error rate values, sequence length and quality score cutoffs?

Thank you for these suggestions. Usually, we don’t give a lot of details in this section in order not to repeat ourselves from other papers, which we cite. This section was corrected according to your recommendations, plus the information about the place of sequencing handling has been added in the Acknowledgements section.

Additionally, did you rarefy your data prior to calculating the alpha diversity indices?

Usually, we don’t think it is necessary to rarefy this type of data. Here we are including alpha diversity without and with rarefaction, and you can see that the outcome is the same. So, we decided to leave data unrarefied in all analyses for consistency.

Alpha without rarefaction:

see attachment

Alpha with rarefaction:

see attachment

On Line 153, it would be helpful to provide a numeric citation index for the vegan package. The indices mentioned from Line 147 to 150 do not align with the reference numbers. For Lines 151 to 155, the descriptions for NMDs and CCA seem somewhat repetitive, and it would be beneficial to condense them for clarity.

Thank you, vegan version was added, citation index was fixed. NMDS and CCA description was indeed repetitive with the intention of explaining the usage of both metrics. According toy your suggestion, we altered this description.

Line160, a typo, should be ‘slightly acidic pH’

Thank you, this mistake has been corrected

Line 201-220, I would suggest to tune down the interpretation of Shannon index.

Starting from Line 228 onwards, I recommend reviewing the journal's italicization requirements for scientific names. Typically, scientific names up to and including the family level are italicized, while names beyond that level are not.

This is an embarrassing mistake from our part. All phyla names have been converted from italics.

Furthermore, consider that the discussion related to various phyla tends to divert from the primary storyline you intend to emphasize. It may be beneficial to restructure this section based on the points highlighted in Lines 273-274 to maintain better focus.

This section has been reorganized.

Regarding Figure 3, kindly include the number of observations for each boxplot. Based on the available information, it appears that the number of observations is limited. If this is the case, statistical significance may not be robust, and it may be advisable to omit it.

The information was added. We had four observations, and you are right, it is not totally correct to use Tukey HSD test in this case. We tried Mann Whitney with Benjamini Hochberg correction (see below). As you can see, significance is not here. In the end we decided settle in the middle and leave in the manuscript a version with Mann Whitney with not adjusted p-values. We admit that our original analysis is a p-value hacking, but we think that it is not drastically incorrect and better highlights the point in the manuscript.

Additionally, we did a richness check via the breakaway package, specialized for this type of data  (Poisson model) - despite the questionable number of repetitions we can say that the alpha diversity is at least different between Plaggen - Mature soil. Confidence intervals are so small that they are inside the richness estimate (little lines inside circles, see plot below).

As for Figure 5, please ensure that the sum of contributions from each axis to the total variation, which can be explained by the environmental variables, does not exceed 100%. Please review your source code for troubleshooting.

Thank you for noticing this. The analysis was redone, and the figure has been changed.

Figure 6, class level (e.g. Alpha- and Gamma-proteobacteria) and phylum level were mixed. If this is what you want to demonstrate, please state in the figure caption.

Yes, usually we like to focus on separate classes from Pseudomonadota. The figure caption was updated accordingly.

Figure 7, mixed condition similar to Fig 6, as your mentioned at genus level, while a lot of were at taxonomic rank of order.

The information about the taxonomic ranking of phylotypes was updated in the caption.

Last, ‘data availability’ about your sequencing data are missing. Typically, sequencing data is uploaded to databases like SRA of NCBI or ENA, and corresponding BioProject ID and sample IDs are provided.

The information about BioProject ID at SRA has been added.

Sincerely yours,

Evgeny Abakumov, corresponding author,

Professor, heap of Department of Applied Ecology,

Saint-Petersburg State University, Russia

Reviewer 2 Report

TITLE: The microbiome of various type of abandoned fallow soils of south taiga (Novgorod region, Russian North-West)

Abstract

L23. Missing “.”

L29. Keywords: “Borovitchy” or “Russia”. No need for two geographical location keywords.

Materials and Methods

No information about the Basal respiration (BRS) and Substrate induces respiration (SIR) and Particle size distribution. However, in Results and Discussion part, there is data about these analyses.

L 81-84. Study sites need to clarify. Geographical distances between study sites would be informative.

L92-116. Missing clear names of study sites. Author use different names, terms and codes throughout the manuscript. For example, in methods, there is names: 1) mature soil as Podzol; 2) 30-year old vegetable garden; 3) secondary forest; 4) pasture digression; 5) hayfield soil; 6) Plaggen. In Table 1, Table 2, Figure 3 – Figure 7 different coding. Please use clear notation to each studied site for better understanding.

L126-129. Please describe in more detail, how the soil samples were taken (depth, time, with soil trill?).

L147-157. New subtitle “Statistical analysis”

Results and discussion

L160. According to results from Table 1 “….characterized by slightly alkaline pH values” -> acidic pH values

Table 1. The table is not uniform. Standard errors or standard deviations shown with only some results.

Table 2. Confusing table. Used only “Sample codes” that are not in a logical order. Sample codes A-F have not been deciphered.

Figure 3. Maybe it would be clearer when results with Tukey HSD test were in table. Significant level p-value <0.0001 has not been used in Figure.

Author Response

Reviewer 2

TITLE: The microbiome of various type of abandoned fallow soils of south taiga (Novgorod region, Russian North-West)

Abstract

L23. Missing “.”

Thank you, corrected.

L29. Keywords: “Borovitchy” or “Russia”. No need for two geographical location keywords.

Thank you, corrected. Geographical name replaced and two additional key words has been added.

Materials and Methods

No information about the Basal respiration (BRS) and Substrate induces respiration (SIR) and Particle size distribution. However, in Results and Discussion part, there is data about these analyses.

Thank you, corrected, methods has bee added.

L 81-84. Study sites need to clarify. Geographical distances between study sites would be informative.

Thank you, corrected, data on sampling plots has been added.

L92-116. Missing clear names of study sites. Author use different names, terms and codes throughout the manuscript. For example, in methods, there is names: 1) mature soil as Podzol; 2) 30-year old vegetable garden; 3) secondary forest; 4) pasture digression; 5) hayfield soil; 6) Plaggen. In Table 1, Table 2, Figure 3 – Figure 7 different coding. Please use clear notation to each studied site for better understanding.

 Thank you, the names of the sites were unified across the manuscript, figures, and tables.

L126-129. Please describe in more detail, how the soil samples were taken (depth, time, with soil trill?).

Thank you, corrected

L147-157. New subtitle “Statistical analysis”

Thank you, corrected

Results and discussion

L160. According to results from Table 1 “….characterized by slightly alkaline pH values” -> acidic pH values

Thank you, of cause, this is mistype. Has been corrected.

Table 1. The table is not uniform. Standard errors or standard deviations shown with only some results.

            Thank you, corrected

Table 2. Confusing table. Used only “Sample codes” that are not in a logical order. Sample codes A-F have not been deciphered.

            Thank you, corrected

Figure 3. Maybe it would be clearer when results with Tukey HSD test were in table. Significant level p-value <0.0001 has not been used in Figure.

Thank you for noticing this. We changed the statistical analyses of alpha diversity according to the recommendations of other reviewers, so the figure and the caption has been updated. In the end we used Mann-Whitney test and not a lot of values remained significant. So we decided that graphical representation of this data is more clear than a table.

Sincerely yours,

Evgeny Abakumov, corresponding author,

Professor, heap of Department of Applied Ecology,

Saint-Petersburg State University, Russia

Reviewer 3 Report

The manuscript showed a Russian case on the diversification of soil microbiome under the effect of six types of land use, and showed some relationship among microbiome, nutrient contents, and soil physical and chemical properties. The whole story is more complete. Some suggestions:

1. Tell the longitude and latitude of the study site’s location in the M&M.

2. “Fallow lands play an important role in the sequestration of carbon dioxide, they can accumulate carbon or initialize its emissions, depending on the type and age of succession, and type of land use.” “Figure 1. The study area. Novgorod region.” These are ungrammatical sentences. Please check the full text.

3. N-NH4 should be N-NH4+; N-NO3 should be N-NO3-.

4. For Table 1, pH should put on first and then TOC. And why some told the error value, some not.

Slightly more language issues exist and need to be improved.

Author Response

Reviewer 3

The manuscript showed a Russian case on the diversification of soil microbiome under the effect of six types of land use, and showed some relationship among microbiome, nutrient contents, and soil physical and chemical properties. The whole story is more complete. Some suggestions:

  1. Tell the longitude and latitude of the study site’s location in the M&M.

Thank you, longitude and latitude has been provided

  1. “Fallow lands play an important role in the sequestration of carbon dioxide, they can accumulate carbon or initialize its emissions, depending on the type and age of succession, and type of land use.” “Figure 1. The study area. Novgorod region.” These are ungrammatical sentences. Please check the full text.

Thank you, corrected.

  1. N-NH4 should be N-NH4+; N-NO3 should be N-NO3-.

Thank you, corrected.

  1. For Table 1, pH should put on first and then TOC. And why some told the error value, some not.

Thank you, corrected.

Sincerely yours,

Evgeny Abakumov, corresponding author,

Professor, heap of Department of Applied Ecology,

Saint-Petersburg State University, Russia

Round 2

Reviewer 1 Report

The long introduction extends until the end without a paragraph break.

Please ensure that the figure captions are as comprehensive as possible, providing readers with essential information without the need to refer to the main text. Avoid captions like 'Figure 1. The study area' and provide sufficient details.

In Table 1, please correct the subscripts and superscripts as needed.

For Figure 3, make sure to include the number of observations for each box in the figure caption.

Regarding Figure 5, please modify the labels for the subscripts of NO3 and NH4, and ensure consistency with the labels within the plot.

Author Response

The long introduction extends until the end without a paragraph break.

Thank you, we added paragraph breaks in the introduction.

Please ensure that the figure captions are as comprehensive as possible, providing readers with essential information without the need to refer to the main text. Avoid captions like 'Figure 1. The study area' and provide sufficient details.

Thank you, figure captions have been updated

In Table 1, please correct the subscripts and superscripts as needed.

Corrected

For Figure 3, make sure to include the number of observations for each box in the figure caption.

It was already mentioned that there are four replicates, but we changed the formulation a bit.

Regarding Figure 5, please modify the labels for the subscripts of NO3 and NH4, and ensure consistency with the labels within the plot.

The figure was modified.

Sincerely yours,

Evgeny Abakumov, corresponding author